# Antimicrobial Efficacy of Phyto-L, Thiosulfonate from *Allium* spp. Containing Supplement, against *Escherichia Coli* Strains from Rabbits

**DOI:** 10.3390/vetsci10070411

**Published:** 2023-06-23

**Authors:** Francesco D’Amico, Gaia Casalino, Francesca Rita Dinardo, Michele Schiavitto, Antonio Camarda, Diana Romito, Antonella Bove, Elena Circella

**Affiliations:** 1Department of Veterinary Medicine, University of Bari “Aldo Moro”, S. P. Casamassima km 3, 70010 Valenzano, BA, Italy; francesco.damico@uniba.it (F.D.); gaia.casalino@uniba.it (G.C.); francesca.dinardo20@gmail.com (F.R.D.); antonio.camarda@uniba.it (A.C.); diana.romito@uniba.it (D.R.); antonella.bove@uniba.it (A.B.); 2Italian Rabbit Breeders Association—ANCI, Contrada Giancola snc, 71030 Volturara Appula, FG, Italy; micheleschiavitto@anci-aia.it

**Keywords:** colibacillosis, rabbit, *Escherichia coli*, organosulfur compounds, antimicrobial efficacy, *Allium*

## Abstract

**Simple Summary:**

The aim of this study was to evaluate the minimum inhibitory concentration (MIC) and minimum bactericidal concentration (MBC) of Phyto-L (Pro Tech s.r.l.), a commercial product containing organosulfur compounds (OSCs) such as propyl propane thiosulfonate (PTSO) from *Allium* spp., on 108 enteropathogenic *E. coli* (EPEC) strains responsible for colibacillosis in rabbits. Bacterial suspensions with a charge of 10^8^ CFU/mL were tested with different concentrations (20, 10, 5, 2.5, 1.25, 0.6, 0.3, and 0.15 μL/mL) of Phyto-L. To evaluate MBC values, bacterial suspensions corresponding to the MIC and above the MIC were plated on Tryptic Soy agar (TSA) without Phyto-L. The MICs of the tested strains corresponded to 1.25 μL/mL (37/108-34.3%), 2.5 μL/mL (70/108-64.8%), and 5 μL/mL (1/108-0.9%). The MBCs were 1.25 μL/mL (15/108-13.9%), 2.5 μL/mL (46/108-42.6%), 5 μL/mL (9/108-8.3%), 10 μL/mL (20/108-18.5%), 20 μL/mL (8/108-7.4%), and higher than 20 μL/mL (10/108-9.3%). Based on the results obtained, Phyto-L has antibacterial activity on EPEC strains. Therefore, in field applications, Phyto-L should be useful in limiting the *E. coli* load in the rabbit gut, preventing the occurrence of colibacillosis. Moreover, considering that 10^4^–10^5^ CFU/g of feces is the charge of *E. coli* normally present in the intestinal contents of rabbits under physiological conditions, it is possible that lower dosages than those found in this study may be effective in preventing the disease in rabbit farms.

**Abstract:**

Colibacillosis, caused by enteropathogenic *Escherichia coli* (EPEC), is one of the most common diseases in rabbit farms, resulting in economic losses due to mortality and decrease in production. Until recently, antimicrobials were used to both treat and prevent disease on livestock farms, leading to the possible risk of antimicrobial resistance (AMR) and the selection of multidrug-resistant (MDR) bacteria. Therefore, interest in alternative control methods, such as the use of natural substances, has increased in the scientific community. The aim of this study was to evaluate the antimicrobial efficacy of Phyto-L (Pro Tech s.r.l.), a product containing organosulfur compounds (OSCs) such as propyl propane thiosulfonate (PTSO) from *Allium* spp., against 108 strains of *E. coli* isolated from rabbits with colibacillosis from 19 farms. The minimum inhibitory concentration (MIC) and minimum bactericidal concentration (MBC) of Phyto-L were assessed. Bacterial suspensions with a charge of 10^8^ CFU/mL, corresponding to those found in the rabbit gut under pathologic conditions, were tested with different concentrations from 20 to 0.15 μL/mL of Phyto-L. For each strain, the MIC and concentrations above the MIC were plated on Tryptic Soy agar (TSA) without Phyto-L to assess the MBCs. MIC and MBC values ranged from 1.25 to 5 μL/mL and 1.25 to 20 μL/mL, respectively, depending on the strain tested. The data showed an interesting antibacterial activity of Phyto-L against EPEC strains. Therefore, this product could be effective in preventing colibacillosis in field application, especially considering that 10^4^–10^5^ CFU/g of feces is the amount of *E. coli* usually found in the gut contents of rabbits under physiological condition.

## 1. Introduction

Enteric and respiratory diseases are the main causes of mortality and production loss in meat rabbitries [1]. Colibacillosis is the most common intestinal disease. The agent of colibacillosis, *Escherichia coli* (*E. coli*), is a Gram-negative, rod-shaped, aerobic and facultative anaerobic, non-spore-forming coliform bacterium. *E. coli* may be responsible for disease in several animal species, including humans, cattle, pigs, sheep, goats, poultry, and rabbits. As in most animal species, *Escherichia coli* (*E. coli*) is a normal component of rabbit digestive flora [2]. The proliferation of *E. coli* in ileocecal contents occurs especially in post-weaning rabbits, causing diarrhea and economic decline due to weight loss and 20–30% mortality [3], which may be higher if associated with certain serotypes of *E. coli* [4]. Clinical signs may vary depending on the severity of the infection, immune status, and age of the rabbit. Colibacillosis usually occurs as acute enteritis in young rabbits and is typically characterized by the sudden onset of diarrhea, which appears watery, yellow, and may contain mucus. In addition, lethargy, dehydration, hypothermia, and anorexia are observed. Death occurs within 24 to 48 h of the onset of clinical signs. In adult rabbits, colibacillosis may evolve into chronic enteritis, and recurrent episodes of diarrhea and weight loss are found. Based on clinical symptoms observed in animals and virulence-associated genes, *E. coli* strains are classified into different pathotypes responsible for intestinal or extra-intestinal syndromes. Notably, the *E. coli* attachment and effacing (AEEC) pathotype includes both enterohemorrhagic (EHEC) and enteropathogenic (EPEC) strains [5,6,7,8,9,10,11]. Enterohemorrhagic (EHEC) strains share two primary virulence factors: the pathogenicity island ‘LEE’ and prophages encoding one or more Shiga toxins [12]. Enteropathogenic *E. coli* (EPEC) strains responsible for colibacillosis in rabbits are among the major diarrheal *E. coli* pathotypes causing attachment and effraction (A/E) lesions, with the classic definition of intimate adhesion and effacement of intestinal microvilli. Their virulence is related to the possession of intima (*eae* gene) [13] and bundle-forming pili (*bfp* gene) [14], (*af/r1* and *af/r2* genes) [15]. Another way to determine the enteropathogenic ity of *E. coli* is to assess serogroup and sugar fermentation ability (biotype), targeted to establishing a link between biotype/serotype and rabbit mortality [3].

Treatment of colibacillosis involves the use of necessary drugs to control the bacterial infection and prevent secondary complications. The most used antibiotics are enrofloxacin and trimethoprim–sulfamethoxazole [16]. The choice of antibiotic should be based on antibiogram results, considering the specific susceptibility level of the bacterial strain detected in the disease outbreak [17]. In addition, monitoring of *E. coli* strains in rabbit populations is important to minimize the risk of selecting resistant strains [18]. Prevention of colibacillosis in rabbit farms is based on implementing biosecurity measures and improving husbandry practices [19]. Considering that colibacillosis is a disease that may be affected by several predisposing factors, it is relevant to ensure suitable ventilation and maintain good environmental hygiene conditions by regularly cleaning and disinfecting cages and equipment, such as feeders and troughs; avoid overcrowding and ensure proper handling of rabbits to minimize stress; and provide adequate nutrition by feeding rabbits balanced diets containing correct amounts of fiber. In addition, an appropriate weaning age for bunnies should be adopted [19]. Vaccine prophylaxis is not as widespread on rabbit farms as on poultry ones, essentially because of the lack of confidence that many farmers have in this method of prevention, owing to the cost of vaccines, the antigenic variability of disease-causing bacteria, and the presence of complex, multifactorial agents that may affect the effectiveness of the intervention [20].

Until recently, antibiotics have been widely used to prevent colibacillosis in the rabbit industry. Considering that for some antibiotics, a positive relationship has been found between antimicrobial consumption and antimicrobial resistance in bacteria isolated from animals and humans, European legislation has drastically restricted the use of antibiotics on livestock for preventive purposes [21]. Therefore, a growing interest in alternative methods of infectious disease control has been observed. The possibility of using natural substances or their derivatives as an alternative to antibiotics has prompted the scientific community to investigate their potential effectiveness [22]. Among them, *Allium* spp. have attracted great interest in human medicine [23] because of their various biological functions, such as anti-inflammatory, antiatherosclerosis, antidiabetic, antimutagenic, anticarcinogenic, antioxidant, and immunomodulatory activities [24,25]. *Allium* spp. also seems to have antimicrobial properties against several pathogens [26,27,28,29]. The pharmacological effects of garlic and onion are related to their organosulfur compounds such as thiosulfonates [30,31], which are responsible for the typical pungent smell and healing properties [32]. When the plant tissues are disrupted, several thiosulfonate compounds containing combinations of allyl, methyl, or propyl groups are produced by enzymatic hydrolysis. Allyl-2propenylthiosulfinate is synthesized from alliin (S-allyl-L-cysteine sulfoxide) and, together with diallyl sulfide and its derivatives, constitutes one of the most important biologically active compounds found in garlic [33,34,35,36,37]. Thiosulfates are composed of sulfur atoms covalently bonded to other sulfur atoms and are unstable compounds that are easily oxidized in air [38], and their biological action appears to be linked to the number of sulfur atoms present [35,39,40]. It is known that through this instability, they participate in the activation and inactivation of enzymes, modify cellular protein activity [41], have a radioprotective effect by removing free radicals, inhibit the growth of tumor cells, and have a detoxifying and anti-aggregating effect on both human and canine platelets [42,43,44]. Concerning their mechanism of action, thiosulfates are able to inhibit the mitochondrial electron transport chain by reducing oxygen consumption and mitochondrial membrane potential and the amount of cellular ATP, resulting in toxicity to yeasts such as Saccharomyces cerevisiae [45]. Diallyl sulfide can penetrate the cell membrane, causing loss of cell integrity and the ability to synthesize ATP and resulting in bacterial lysis, inactivation of metabolic proteins, and inhibition of protein synthesis in both *Campylobacter jejuni* and *Helicobacter pilori* [46,47]. It is also capable of inhibiting the growth of *Klebsiella pneumoniae*, *Salmonella typhimurium*, and *Helicobacter pylori* by interfering with the activity of enzymes, including the arylamine N-acetyltransferase necessary to keep the bacterial cell metabolically active and therefore alive [39,48].

In our study, we tested the antimicrobial efficacy of a commercial product (Phyto-L) containing OSCs as thiosulfonates from *Allium* spp. against 108 strains of *E. coli* isolated from rabbits with colibacillosis from 19 farms. In order to provide a scientific basis for effective application in the rabbit industry, the minimum inhibitory concentration (MIC) and minimum bactericidal concentration (MBC) of Phyto-L were assessed.

## 2. Materials and Methods

### 2.1. Organosulfur Compounds

Phyto-L, supplied by Pro Tech Animal Nutrition s.r.l. (Via Zerbi, 47, Carbonara Scrivia, AL, Italy), was used to test antimicrobial efficacy. This product contains organosulfur compounds such as PTSO at a concentration of 170,000 µg/mL, supported on an inert carrier (glyceryl polyethylene glycol ricinoleate-E-484).

### 2.2. Bacterial Strains Used for the Study

The study was carried out in vitro on 108 *E. coli* strains from rabbits. All strains were previously collected in a period ranging from 2006 to 2023 and stored at −20 °C in Brucella broth and glycerol (10%) in the bacterial collection of the Avian Diseases Unit of the Department of Veterinary Medicine (DVM), University of Bari, Italy. The strains were previously isolated from the cecum contents of weaned rabbits that died from enteritis at ages ranging from 32 to 60 days. Rabbits were from 19 intensive rabbit farms in central and southern Italy. *E. coli* strains were identified as EPEC based on the detection of *eae* and *afr2* genes by polymerase chain reaction (PCR) according to protocols already described [15,17]. All strains were grown on tryptic soy agar (TSA) (OXOID, Basingstoke, UK) at 37 °C overnight before performing the analyses.

### 2.3. Determination of Minimal Inhibitory Concentrations (MIC)

According to CLSI standards [49], a 0.5 McFarland suspension corresponding to 1–2 × 10^8^ CFU/mL was prepared for each strain. Mueller Hinton Broth (Oxoid) was prepared, reconstituted according to the manufacturer’s instructions, and autoclaved at 121 °C for 15 min. The broth was brought to a temperature of 50 °C in a thermostatic bath. Phyto-L, previously diluted in DMSO in the ratio of 9:1 (9 parts of Phyto-L for one part of DMSO) according to [50,51,52], was added to the broth at concentrations ranging from 0.15 to 20 μL/mL (0.15, 0.3, 0.6, 1.25, 2.5, 5, 10, and 20 μL/mL). The broths were dispensed into 0.6 mL Eppendorf tubes, aliquoting 100 μL per tube.

Ten microliters of each bacterial suspension (1–2 × 10^8^ CFU/mL) was added to the Eppendorf tubes containing broths with different concentrations of Phyto-L. As a positive control of bacterial growth, 10 μL of each bacterial suspension was inoculated into the broth containing DMSO at the highest concentration (0.2% DMSO) used in the trials. The contents of the Eppendorf tubes were mixed by automated vortexing and incubated at 37 °C for 24 h under aerobic conditions. Inhibition or growth of bacteria was interpreted according to the clarity or turbidity of the inoculated broths, respectively. The MIC was identified as the minimum concentration of Phyto-L at which the broth appeared clear. Each experiment was carried out twice.

### 2.4. Determination of Minimal Bactericidal Concentrations (MBC)

MBC determination was performed according to NCCLS [53], with some modifications. In detail, Petri plates containing Mueller Hinton agar (Oxoid), previously prepared according to the manufacturer’s instructions and autoclaved at 121 °C for 15 min, were used. For each strain, the entire volume of broth corresponding to the MIC and all concentrations higher than the MIC were inoculated on Mueller Hinton agar.

All plates were incubated at 37 °C for 24 h under aerobic conditions. The MBC was identified as the lowest concentration at which no bacterial growth was observed on the culture medium.

## 3. Results

The MIC corresponded to 2.5 μL/mL for 70 out of 108 (64.8%) strains (Table 1). The MIC was 1.25 μL/mL for 37 (34.3%) strains and 5 μL/mL for one strain.

The MBC was 1.25 and 2.5 μL/mL for 15 (13.9%) and 46 (42.6%) strains, respectively. It resulted in 10 μL/mL for 20 (18.5%) strains. The MBC was 20 μL/mL for 8 (7.4%) strains and higher than 20 μL/mL for 10 (9.3%) strains.

Considering the rabbit farms, the MIC corresponded to 1.25 μL/mL for strains identified in farms 3, 13, 14, 16, and 17 and to 2.5 μL/mL in farms 4, 5, 7, and 10 (Table 2). Conversely, MIC values were 1.25 or 2.5 μL/mL in farms 1, 2, 6, 8, 11. The MIC corresponded to 5 μL/mL only for a strain from farm 1.

In certain cases, such as farms 8, 10, 14, and 16, similar MBC and MIC values were observed. As expected, the susceptibility to the product varied depending on the strain isolated from each farm.

The highest variability in susceptibility to the product was found when testing strains from farm 1 (Table 3).

## 4. Discussion

Based on the results of this study, Phyto-L effectively inhibited the growth of *E. coli* in rabbits, with MIC values ranging from 1.25 to 5 μL/mL, depending on the strain tested. This finding is of interest because the strains were tested in a bacterial load of 10^8^ UFC/mL, which is the amount of *E. coli* detected per g of feces in rabbits affected by colibacillosis [54], whereas 10^4^ or 10^5^ UFC/mL/g of feces is normally found in the intestinal contents of rabbits under physiological conditions [55]. A previous study performed using garlic against Salmonella evidenced that lower concentrations of garlic were inhibitory against lower loads of bacteria [26]. In addition, the gut microbiota limits the proliferation of *E. coli* in rabbits under physiological conditions [56]. The immunomodulatory effect of *Allium* spp. also improves the activity of the intestinal microbiota as well as the production parameters of rabbits [57,58]. It is therefore very likely that when combined with the action of the intestinal microflora, the efficacy of Phyto-L in the prevention of colibacillosis in rabbit flocks may increase, leading to the use of lower dosages under field conditions than those suggested by the MIC values found in our study.

In addition, Phyto-L showed bactericidal effects on *E. coli* strains from rabbits with MBC values ranging from 1.25 up to 20 μL/mL, which was in accordance with other studies in vitro [59,60]. This bactericidal effect indicated that the product could be effective not only in the prevention but also in the treatment of colibacillosis when used at higher doses. The bactericidal effect may depend on the structural characteristics of various microorganisms that influence their susceptibility to *Allium* components [60,61,62,63,64,65,66,67].

Concerning the inhibitory efficacy of Phyto-L found on strains tested in vitro, similar results have been observed in other studies using *Allium* spp. against various bacteria [26,68,69,70]. MIC values obtained for *E. coli*, *Staphylococcus aureus*, *Pseudomonas* and *Salmonella enterica* serovar Typhi [68], *Streptococcus mutans* [69], and *Salmonella* e. sub e. ser. Enteritidis [26,71] ranged from 0.02 to 6.25 mg/mL of garlic, depending on the bacterial species. Another study showed the antimicrobial efficacy of garlic against *S. aureus* if garlic was in concentrations above 7.50 mg/mL [59]. In addition to the bacterial species and bacterial load tested, variability in efficacy values may also be due to the laboratory methods used for investigation and the treatment of the natural substance before testing, which may affect the stability of allicin [72], the main active ingredient in garlic. Other organosulfur compounds such as PTSO have demonstrated a significant antimicrobial activity against multidrug-resistant isolates of *E. coli* and other *Enterobacteriaceae* spp. with MIC values in the range of 64–128 µg/mL [73]. In addition, the same authors demonstrated a higher activity against Gram-positive bacteria such as *S. aureus* and reported the antimicrobial activity of PTSO via the gas phase [74]. Other previous studies have reported the in vitro bactericidal activity of thiosulfinates against *E. coli* and *Salmonella typhimurium* in pig feces [75]. Recently, the antibacterial activity of PTSO was also described against fish pathogens [76].

In our study, the inhibitory efficacy of Phyto-L varied according to the strain tested, even when identified in rabbits from the same herd. This finding agreed with another study performed on *E. coli* and *S. aureus* strains, where similar MIC values were shown between the two different bacterial species, whereas MIC variability from 4 to 8 mg/mL of garlic was found within each species, depending on the strain [70].

Natural substances usually show efficacy at dosages higher than those referred to for antibiotic molecules and, in addition, require longer administration times. *Allium* spp. has been found to be effective in the treatment and prevention of *E. coli* infections in chickens [63]. In broilers, administration results in a reduction of intestinal coliforms, as well as improved production performance [77]. A study of chickens reported the antimicrobial efficacy of garlic after administration for 56 days [78]. In addition, other in vivo studies in broilers fed with similar compounds found antimicrobial activity against *E. coli* and Salmonella. Furthermore, an improvement in body weight was observed in animals fed a diet with an *Allium* product [57]. A comparative study in rabbit farms evaluated the antimicrobial efficacy of garlic and florfenicol (FFC), a broad-spectrum antibiotic, against *E. coli* serotype O55:H7 [79,80]. Separate groups of rabbits were administered FFC for 5 days and garlic for 14 days, starting 7 days before the challenge infection and up to 7 days after infection. Compared with the control group, a reduction in symptoms and mortality was observed in both treated groups, as well as the maintenance of better productive performance and a reduction in fecal excretion of the *E. coli* strain used for infection. However, higher interferon-gamma (IFN-γ) and phagocyte levels were found in the garlic-treated group.

Moreover, the efficacy of Phyto-L could be enhanced by its combination with other natural products. In vitro, thyme, peppermint, sage, black pepper, and garlic showed a greater antimicrobial effect against *Bacillus subtilis* and *Salmonella* Enteritidis when combined rather than analyzed individually [81]. A mixture consisting of organic acids and cinnamon administered through the feed in turkey flocks resulted in the reduction of lesions induced by an antibiotic-resistant strain of *E. coli* 078 and a reduction in the intestinal concentration of the germ [82].

In any case, the use of natural substances could be a viable alternative to the use of antibiotics, especially on rabbit farms, where antimicrobial consumption (ACM) is the highest among food-producing animals [83]. The widespread use of antibiotics to prevent infectious diseases in animals increases the risk of antimicrobial resistance [84,85], and medicated feed containing antibiotics, widely used in the past, may have contributed to the selection of antibiotic-resistant bacterial populations in the environment and in animals [86]. More recently, antimicrobial consumption in rabbit breeding decreased with some drugs, although the use of substances such as fluoroquinolones increased [21]. Minimizing antibiotic use and finding alternative strategies for infection control are essential steps for reducing antimicrobial resistance. Therefore, in line with the European directives, Regulation (EU) 2019/6 and the circular n. 1/2022 on the guidelines for the prudent use of antibiotics in breeding rabbits for meat production, it is useful to increase the use of natural substances. Considering that colibacillosis is affected by several environmental and management factors, the association between the proper application of biosecurity and hygiene measures in rabbit farm management and the administration of natural substances is very relevant.

## Figures and Tables

**Table 1 vetsci-10-00411-t001:** MIC and MBC of Phyto-L found for 108 *E. coli* strains.

Phyto-L μL/mL (OSCs mg/mL)	MIC N° of Strains (%)	MBC N° of Strains (%)
>20 (>3.4)	0 (0)	10 (9.3)
20 (3.4)	0 (0)	8 (7.4)
10 (1.7)	0 (0)	20 (18.5)
5 (0.85)	1 (0.9)	9 (8.3)
2.5 (0.425)	70 (64.8)	46 (42.6)
1.25 (0.2125)	37 (34.3)	15 (13.9)
0.6 (0.102)	0 (0)	N.D. *
0.3 (0.051)	0 (0)	N.D.
0.15 (0.0255)	0 (0)	N.D.

* ND: not determined.

**Table 2 vetsci-10-00411-t002:** MIC and MBC of Phyto-L found for *E. coli* strains identified in the different farms.

	MIC						MBC					
	Phyto-L Concentrations (μL/mL)						Phyto-L Concentrations (μL/mL)					
	>20	20	10	5	2.5	1.25	>20	20	10	5	2.5	1.25
Farm (N° of Tested Strains)	N° of Strains (%)						N° of Strains (%)					
1 (25)	0 (0)	0 (0)	0 (0)	1 (4)	9 (36)	15 (60)	9 (36)	1 (4)	0 (0)	7 (28)	5 (20)	3 (12)
2 (26)	0 (0)	0 (0)	0 (0)	0 (0)	21 (80.8)	5 (19.2)	0 (0)	0 (0)	5 (19.2)	0 (0)	17 (65.4)	4 (15.4)
3 (3)	0 (0)	0 (0)	0 (0)	0 (0)	0 (0)	3 (100)	0 (0)	0 (0)	3 (100)	0 (0)	0 (0)	0 (0)
4 (8)	0 (0)	0 (0)	0 (0)	0 (0)	8 (100)	0 (0)	0 (0)	1 (12.5)	5 (62.5)	0 (0)	2 (25)	ND *
5 (6)	0 (0)	0 (0)	0 (0)	0 (0)	6 (100)	0 (0)	0 (0)	0 (0)	3 (50)	0 (0)	3 (50)	ND
6 (7)	0 (0)	0 (0)	0 (0)	0 (0)	5 (71.4)	2 (28.6)	0 (0)	2 (28.6)	1 (14.3)	0 (0)	2 (28.6)	2 (28.6)
7 (2)	0 (0)	0 (0)	0 (0)	0 (0)	2 (100)	0 (0)	0 (0)	2 (100)	0 (0)	0 (0)	0 (0)	ND
8 (13)	0 (0)	0 (0)	0 (0)	0 (0)	11 (84.6)	2 (15.4)	0 (0)	0 (0)	0 (0)	0 (0)	11 (84.6)	2 (15.4)
10 (3)	0 (0)	0 (0)	0 (0)	0 (0)	3 (100)	0 (0)	0 (0)	0 (0)	0 (0)	0 (0)	3 (100)	ND
11 (2)	0 (0)	0 (0)	0 (0)	0 (0)	1 (50)	1 (50)	0 (0)	2 (100)	0 (0)	0 (0)	0 (0)	0 (0)
13 (2)	0 (0)	0 (0)	0 (0)	0 (0)	0 (0)	2 (100)	1 (50)	0 (0)	1 (50)	0 (0)	0 (0)	0 (0)
14 (2)	0 (0)	0 (0)	0 (0)	0 (0)	0 (0)	2 (100)	0 (0)	0 (0)	0 (0)	0 (0)	0 (0)	2 (100)
16 (2)	0 (0)	0 (0)	0 (0)	0 (0)	0 (0)	2 (100)	0 (0)	0 (0)	0 (0)	0 (0)	0 (0)	2 (100)
17 (2)	0 (0)	0 (0)	0 (0)	0 (0)	0 (0)	2 (100)	0 (0)	2 (100)	0 (0)	0 (0)	0 (0)	0 (0)

* ND: Not determined.

**Table 3 vetsci-10-00411-t003:** MIC and MBC values found for each *E. coli* strain tested from farm 1.

	MIC						MBC					
	Phyto-L Concentrations (μL/mL)						Phyto-L Concentrations (μL/mL)					
Strain	>20	20	10	5	2.5	1.25	>20	20	10	5	2.5	1.25
1	-	-	-	-	-	+	-	-	-	+	-	-
2	-	-	-	-	-	+	-	-	-	-	+	-
3	-	-	-	-	-	+	-	-	-	-	-	+
4	-	-	-	-	-	+	-	-	-	-	-	+
5	-	-	-	-	-	+	-	-	-	-	-	+
6	-	-	-	-	-	+	-	-	-	+	-	-
7	-	-	-	-	-	+	-	-	-	-	+	-
8	-	-	-	-	+	-	+	-	-	-	-	-
9	-	-	-	-	+	-	+	-	-	-	-	-
10	-	-	-	-	-	+	+	-	-	-	-	-
11	-	-	-	-	+	-	-	-	-	+	-	-
12	-	-	-	-	+	-	+	-	-	-	-	-
13	-	-	-	-	-	+	-	-	-	-	+	-
14	-	-	-	-	-	+	-	-	-	+	-	-
15	-	-	-	-	+	-	-	-	-	+	-	-
16	-	-	-	-	+	-	-	-	-	-	+	-
17	-	-	-	+	-	-	+	-	-	-	-	-
18	-	-	-	-	+	-	-	+	-	-	-	-
19	-	-	-	-	-	+	-	-	-	+	-	-
20	-	-	-	-	+	-	+	-	-	-	-	-
21	-	-	-	-	-	+	+	-	-	-	-	-
22	-	-	-	-	-	+	-	-	-	-	+	-
23	-	-	-	-	-	+	+	-	-	-	-	-
24	-	-	-	-	-	+	+	-	-	-	-	-
25	-	-	-	-	+	-	-	-	-	+	-	-

## Data Availability

No new data were created or analyzed in this study. Data sharing is not applicable to this article.

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
