# Peer review of "Antimicrobial Efficacy of Phyto-L, Thiosulfonate from Allium spp. Containing Supplement, against Escherichia Coli Strains from Rabbits"

_vetsci, 2023, doi:10.3390/vetsci10070411_

Round 1

Reviewer 1 Report

Antimicrobial efficacy of allyl-sulfide against Escherichia coli strains from rabbits

D’amico et al

Colibacillosis is problematic in animal-based agriculture, leading to weight loss and mortality. Legislation-driven shifts away from antibiotic use in livestock for preventative purposes has led to an urgent need to find alternatives, in particular natural alternatives. A wide range of alternatives have been proposed and tested, but as yet there is no clear winner. In part this arises due to strain variations in naturally occurring colibacillosis causing bacterial populations.

In this study D’amico and co-workers examine the effect of Phyto-L on 108 strains of E.coli isolated from rabbits with colibacillosis from 19 farms. The MIC and MBC were assessed.

Overall the manuscript is well written, but some points need to be addressed.

1) The title, summary and discussion are potentially incorrect and/or misleading. The material tested was Phyto-L not allyl-sulfide. No evidence is presented that allyl-sulfide is the causative agent. This needs to be amended throughout the manuscript e.g. for the title Antimicrobial efficacy of Phyto-L, an allyl-sulfide containing food supplement, against Escherichia coli strains from rabbits

2) No details are given anywhere of what Phyto-L is and the product does not appear to be on the website of the company that supplied it. Similarly, no data (or citation) is given to show how much allyl sulfide is present in Phyto-L or the method by which this was determined.

3) No data is presented that the main effect of Phyto-L is bacteriostatic rather than bacteriocidal (citations regarding garlic components do not present evidence that this is how this mix acts – unless Phyto-L is pure garlic / garlic extract

4) The presentation of the data in table 1 is both difficult to read and loses part of the data collected. I would suggest the authors consider having figure(s) with MIC vs MBC plotted so that for each strain this can be seen. Use of different colours / data point types (squares vs circles vs crosses etc) for different farms (plus maybe some sub panels) could allow farm specific data also to be seen.

Minor points:

Line 144, it would be good to add a couple of words to say how eae and afr2 were detected to save the reader having to read the papers cited

Line 161, should it be “vortexing” and not “vertexing”

Line 161, it may be good to add a couple of words to say how aerobic conditions were maintained in eppendorfs

Section 2.4, the highest concentration of Phyto-L adds 10% DMSO to the culture media. Were controls done to ensure this had no effect? If not then perhaps appropriate citation(s) could be added

Author Response

Dear Reviewer 1,

thank you for your review. The manuscript has been modified according to your comments and suggestions of the other reviewer. Below, I reported the replies step by step. The changes according to your comments are coloured in pale blue in the manuscript. References were renumbered in the manuscript.

Best regards

Elena Circella

Colibacillosis is problematic in animal-based agriculture, leading to weight loss and mortality. Legislation-driven shifts away from antibiotic use in livestock for preventative purposes has led to an urgent need to find alternatives, in particular natural alternatives. A wide range of alternatives have been proposed and tested, but as yet there is no clear winner. In part this arises due to strain variations in naturally occurring colibacillosis causing bacterial populations.

In this study D’amico and co-workers examine the effect of Phyto-L on 108 strains of E.coli isolated from rabbits with colibacillosis from 19 farms. The MIC and MBC were assessed.

Overall the manuscript is well written, but some points need to be addressed.

Comment. 1) The title, summary and discussion are potentially incorrect and/or misleading. The material tested was Phyto-L not allyl-sulfide. No evidence is presented that allyl-sulfide is the causative agent. This needs to be amended throughout the manuscript e.g. for the title Antimicrobial efficacy of Phyto-L, an allyl-sulfide containing food supplement, against Escherichia coli strains from rabbits

Reply. The title and the manuscript have been changed according to your comment, suggestions provided by the other reviewers and additional information received by the product manufacturer.

2) No details are given anywhere of what Phyto-L is and the product does not appear to be on the website of the company that supplied it. Similarly, no data (or citation) is given to show how much allyl sulfide is present in Phyto-L or the method by which this was determined.

Thank you for this observation. We contact the manufacturer of the product that we tested, and they gave us additional information. Phyto-L is the brand name of a product manufactured by DOMCA and commercialized by Pro-tech. According to reviewer 2, we asked for additional information about the ingredients. This product icontains organosulfur compounds (OSCs) such thiosulfonates (which include allyl sulphide), at concentration of 17%, from Allium spp. These active ingredients are supported on an inert carrier (glyceryl polyethylene glycol ricinoleate (E-484). We have included these details in the manuscript. We provided data about the concentration of OSCs in table 1 more correctly. In addition, we discussed the activities of thiosulfonates in the paper, providing new references.

3) No data is presented that the main effect of Phyto-L is bacteriostatic rather than bacteriocidal (citations regarding garlic components do not present evidence that this is how this mix acts – unless Phyto-L is pure garlic / garlic extract

Thank you for your comment. As mentioned before, the product is rich in thiosulfonates. These compounds have been demonstrated to present also bactericidal effect in previous studies both in vitro and in vivo trials in several target species. We have included in the text some of these references about their bactericidal activity.

4) The presentation of the data in table 1 is both difficult to read and loses part of the data collected. I would suggest the authors consider having figure(s) with MIC vs MBC plotted so that for each strain this can be seen. Use of different colours / data point types (squares vs circles vs crosses etc) for different farms (plus maybe some sub panels) could allow farm specific data also to be seen.

Reply. Thank you for this suggestion. It is very interesting to evidence MIC vs MBC for each tested strain but, at the same time, the inclusion of a table reporting data for 108 strains in the manuscript is very difficult. Anyway, according to you suggestion, we add table 3 that reports MIC vs MBC values for each strain from farm 1 in which the highest variability of susceptibility to the product was found.

Minor points:

Comment. Line 144, it would be good to add a couple of words to say how eae and afr2 were detected to save the reader having to read the papers cited

Reply. The method used for the detection of genes has been added. The details of the protocols have not been included because they are already described, and we applied them without modifications.

Comment. Line 161, should it be “vortexing” and not “vertexing”

Reply. It has been corrected.

Comment. Line 161, it may be good to add a couple of words to say how aerobic conditions were maintained in eppendorfs

Reply. No special measures except for a simple aerobic incubator were used to maintain aerobic conditions inside the Eppendorfs, considering that the volume of broth (100 µL) was put inside Eppendorf with a total volume of 600 µL (1 part of broth with 5 parts of air).

Comment. Section 2.4, the highest concentration of Phyto-L adds 10% DMSO to the culture media. Were controls done to ensure this had no effect? If not then perhaps appropriate citation(s) could be added

Reply. Thank you for your comment that gave us the opportunity to explain the issue. Phyto-L was diluted in DMSO in the ratio of 9:1 (i.e. 9 parts of Phyto-L for one part of DMSO) and added in different concentrations to the broths. Therefore, the highest concentration of Phyto L which was used in the study (20 microliter/mL of mixture corresponding to 2%) adds 0.2 % DMSO (and 1.8% of Phyto-L) to the culture media. The protocol for dilute Phyto-L was used according to previous studies on antimicrobial efficacy of essential oils. References have been reported in the manuscript. Broths containing 0.2% (as the highest dilution of DMSO) were inoculated with each bacterial suspension and used as positive control of bacterial growth. It has been specified in the text of manuscript and is tracked in purple, according to suggestion of reviewer 2 also.

Reviewer 2 Report

In this study, the authors aimed to assess the antimicrobial effectiveness of Phyto-L (Pro Tech s.r.l.), a commercial product containing allyl sulfide, a bioactive component of garlic, against 108 strains of enteropathogenic E. coli (EPEC) isolated from rabbits with colibacillosis. Bacterial suspensions with a concentration of 108 CFU/ml, representative of pathologic conditions in rabbit gut, were prepared, and the Minimum Inhibitory Concentration (MIC) and Minimum Bactericidal Concentration (MBC) of Phyto-L were determined. The results demonstrated MIC and MBC values ranging from 1.25 μL/mL to 5 μL/mL and 1.25 μL/mL to 20 μL/mL, respectively. Overall, the commercial product showed predominantly bacteriostatic rather than bactericidal activity.

The study topic is intriguing and relevant, considering the low concentration of active ingredient required for efficacy (μg/ml). However, the title focuses on the antimicrobial efficacy of Allyl-Sulfide, while the authors investigated and discussed the activity of the complete commercial product, Phyto-L, without providing details about other ingredients. This lack of information creates confusion regarding the effects observed, whether attributed solely to the molecule (Allyl-Sulfide) or the entire product (Phyto-L) along with unknown ingredients. Furthermore, considering the well-known antibacterial activity of DMSO, the protocol used to determine the MIC does not mention the use of a negative control involving the appropriate dilution of DMSO. Without addressing these issues, the work cannot be published.

The English language can be marginally improved.

Author Response

Dear Reviewer 2,

thank you for your comments. The manuscript has been modified to your comments and suggestion provided by the other reviewers. Below, I reported the replies to your suggestions. The changes according to your comments are coloured in purple in the manuscript. References were renumbered in the manuscript, according to the changes.

Best regards

Elena Circella

In this study, the authors aimed to assess the antimicrobial effectiveness of Phyto-L (Pro Tech s.r.l.), a commercial product containing allyl sulfide, a bioactive component of garlic, against 108 strains of enteropathogenic E. coli (EPEC) isolated from rabbits with colibacillosis. Bacterial suspensions with a concentration of 108 CFU/ml, representative of pathologic conditions in rabbit gut, were prepared, and the Minimum Inhibitory Concentration (MIC) and Minimum Bactericidal Concentration (MBC) of Phyto-L were determined. The results demonstrated MIC and MBC values ranging from 1.25 μL/mL to 5 μL/mL and 1.25 μL/mL to 20 μL/mL, respectively. Overall, the commercial product showed predominantly bacteriostatic rather than bactericidal activity.

Comment. The study topic is intriguing and relevant, considering the low concentration of active ingredient required for efficacy (μg/ml). However, the title focuses on the antimicrobial efficacy of Allyl-Sulfide, while the authors investigated and discussed the activity of the complete commercial product, Phyto-L, without providing details about other ingredients. This lack of information creates confusion regarding the effects observed, whether attributed solely to the molecule (Allyl-Sulfide) or the entire product (Phyto-L) along with unknown ingredients.

Reply. Thank you very much for this comment. According to your suggestion, we contacted the manufacturer of the product that we tested, and they gave us additional information. Phyto-L is the brand name of a product manufactured by DOMCA and commercialized by Pro-tech. We asked for additional information about the ingredients. This product contains organosulfur compounds (OSCs) such thiosulfonates (which include allyl sulphide), at concentration of 17%, from Allium spp. These active ingredients are supported on an inert carrier (glyceryl polyethylene glycol ricinoleate (E-484). We have included these details in the manuscript. We provided data about the concentration of OSCs in table 1 more correctly. In addition, we discussed the activities of thiosulfonates in the paper, providing new references.

Comment. Furthermore, considering the well-known antibacterial activity of DMSO, the protocol used to determine the MIC does not mention the use of a negative control involving the appropriate dilution of DMSO. Without addressing these issues, the work cannot be published.

Reply. Thank you for your comment. DMSO was used according to previous studies on antimicrobial efficacy of essential oils, which have been reported in the text of manuscript. As now specified in the manuscript, as positive control of bacterial growth, the bacterial suspension from each strain was inoculated in the broth containing only DMSO at the highest dilution (0.18% DMSO) used in the experiments, without Phyto-L.

Reviewer 3 Report

Major comments:

1/ Introduction could be shortened by one-quarter. For example, some data on Italian rabbit farming and on non-antimicrobial action of garlic could be omitted. Instead, the authors should focus more on mechanisms of antimicrobial action of allyl sulfide.

2/ The design of tables should be improved; e.g. the top line in Table 1 is not clear, No is not "a number", Table 2 is not legible and easy to read. In Table 2, the total number strains is 103, not 108, and farms 9, 12 and 15 are not included - why?

3/ What could be a possible explanation for the fact that E. coli strains displayed so great differences in MBC values? For some strains, MBCs of allyl sulfide were ten times larger than for the other. The authors should try to elucidate this phenomenon considering the mode of action of the compound tested. Were the tests repeated to confirm different susceptibility of E. coli strains?

4/ Bacterial concentration of 108 (corresponding to 0.5 of McFarland standard) has been used for making a standardized suspension for many tests. The fact that a specific concentration of a given substance exerts an antimicrobial activity against that number does not indicate that with a lower load of bacteria, lower concentrations will be inhibitory! Thus, the sentences in lines 23-25 and 41-43 are not justified. 

Minor comments:

L.14: 108, not 108 (similarly, L. 23)

L. 65: should be "facultative anaerobic"

L. 120: The sentence needs some attention

L. 166-169: What medium was used for the determination of MBC? Mueller-Hinton or Tryptic Soy Agar?

L. 181: 1.25 uL/mL is not "a rate of frequency"?

L. 189: "3-4 concentrations higher"?

L. 218 (2x): "Salmonella e. sub e." is not correct. What is "Tiphy"?

L. 246: "Enteritidis" must not be italicized

L. 258 (2x): "molecules" should be relaced by "agents", "substances" or other words

L. 284: "blocks"

Some corrections of English are necessary.

Author Response

Dear Reviewer 3,

thank you for your review. The manuscript has been modified according to your comments and suggestions of the other reviewers. Below, I reported the replies, step by step. The changes according to your comments are coloured in yellow in the manuscript. References were renumbered in the manuscript, according to the changes in the paper.

Best regards

Elena Circella

Major comments:

1/ Introduction could be shortened by one-quarter. For example, some data on Italian rabbit farming and on non-antimicrobial action of garlic could be omitted. Instead, the authors should focus more on mechanisms of antimicrobial action of allyl sulfide.

Reply. According to your suggestion, the introduction has been shortened deleting parts concerning rabbit farming and non-antimicrobial action on Allium. Information about the active ingredients of the product have been added. In addition, according to the requests of reviewers 1 and 2, we received additional information about the ingredients of the product by the manufacturer. They explain us that Phyto L is the brand name of a product manufactured by DOMCA and commercialized by Pro-tech. This product contains organosulfur compounds (OSCs) such as thiosulfonates (which include allyl sulphide) at concentration of 17%, derived from Allium spp Therefore, we provided data about the concentration of OSCs in table 1 more correctly. In addition, we discussed the activities of thiosulfonates in the paper, providing new references.

Comment. 2/ The design of tables should be improved; e.g. the top line in Table 1 is not clear, No is not "a number". Table 2 is not legible and easy to read.

Reply. The legend of table 1 has been improved to make clear the top line of the table. According to your suggestion, Table 2 was replaced by a new table, in which data have been reorganized leading to a clearer and easier table.

Comment. In Table 2, the total number strains is 103, not 108, and farms 9, 12 and 15 are not included - why?

Reply. Only one strain was tested in farms 9, 12 and 15. Therefore, strains from those farms are not included in table 2 where the variability of strains susceptibility inside the same farm is reported.

Comment. 3/ What could be a possible explanation for the fact that E. coli strains displayed so great differences in MBC values? For some strains, MBCs of allyl sulfide were ten times larger than for the other. The authors should try to elucidate this phenomenon considering the mode of action of the compound tested. Were the tests repeated to confirm different susceptibility of E. coli strains?

Reply. The tests were performed twice. It has been now specified in the text of the manuscript. The reason for MBC sometimes larger several times than MIC is not clear, but it was probably due to the lower bactericidal efficacy. We think that further studies will be probably useful to give an explanation.

Comment. 4/ Bacterial concentration of 108 (corresponding to 0.5 of McFarland standard) has been used for making a standardized suspension for many tests. The fact that a specific concentration of a given substance exerts an antimicrobial activity against that number does not indicate that with a lower load of bacteria, lower concentrations will be inhibitory! Thus, the sentences in lines 23-25 and 41-43 are not justified.

Reply. Yes, this is only a hypothesis, based on the results of a previous study about the antimicrobial efficacy of garlic against Salmonella Enteritidis strains which were tested in two different bacterial charges, 106 UFC/ml and 104 UFC/ml (Circella E., Casalino G., D’Amico F., Pugliese N., Dimuccio M.M., Camarda A., Bozzo G. In Vitro Antimicrobial Effectiveness Tests Using Garlic (Allium sativum) against Salmonella enterica Subspecies enterica Serovar Enteritidis. Antibiotics 2022, 11, 1481. https://doi.org/10.3390/antibiotics11111481). The text of the manuscript has been modified according to your suggestion.

Minor comments:

Comment. L.14: 108, not 108 (similarly, L. 23)

Reply. It has been corrected.

Comment. L. 65: should be "facultative anaerobic"

Reply. It has been corrected.

Comment. L. 120: The sentence needs some attention

Reply. The sentence has been corrected.

  1. 166-169: What medium was used for the determination of MBC? Mueller-Hinton or Tryptic Soy Agar?

Reply. Mueller Hinton agar was used for the determination of MBC. The mistake has been correct in the text. Instead, Triptic Soy Agar was used to culture the strains before the preparation of bacterial suspensions (paragraph 2.3)

Comment. L. 181: 1.25 uL/mL is not "a rate of frequency"?

Reply. Yes, you are right, the sentence has been corrected.

Comment. L. 189: "3-4 concentrations higher"?

Reply. A more correct word has been used in the sentence.

Comment. L. 218 (2x): "Salmonella e. sub e." is not correct. What is "Tiphy"?

Reply. The correct name of Serovar Typhi has been reported in the text of the manuscript.

Comment. L. 246: "Enteritidis" must not be italicized

Reply. The italic has been removed.

Comment. L. 258 (2x): "molecules" should be relaced by "agents", "substances" or other words

Reply. Molecules has been replaced by drugs and substances.

Comment. L. 284: "blocks"

Reply. It has been corrected.

Reviewer 4 Report

This is the review of the manuscript “Antimicrobial Efficacy Of Allyl-Sulfide Against Escherichia Coli Strains From Rabbits “.

In general the use of bioactive components against infectious agents for human and animals are in center of research for many reasons. Unfortunately this manuscript has many flaws and it is written not with the proper attention.

To start with author’s credentials should be written in English, there must be an English translation of the 2nd institute.

Simple summary and abstract, which is the 1st readers will check are very badly written, English editing and many mistakes, not clear scientific meanings and controversial results.

In Introduction, very extensive introduction, to some points just narrative without any benefit and no use of references, i.e line 90-107.

In lines 68-70, the use of 1989 reference is quite old to talk about 20-30% losses plus, the reference used is not talking about due to E.coli the losses as authors claim.

In materials and methods section, authors claim that from 19 farms were taken samples of rabbits died from enteritis, without any other data of farms and no strains presented and no data of rabbits died i.e ages, weaned or not weaned at least.

In 2.4 paragraph, no references was presented for the method used for MIC and paragraph 2.5 for MBC.

The presence of table 2. In the results section, confuses readers as suddenly from nowhere we have a discussion about farms and strains but no mention of these was made earlier.

In discussion paragraph there many problems regarding authors discussion about their results and the references used to back authors claims.

Line 201, ref. 35 is not supporting authors writing, also line 202, ref 36.

Line 205, ref. 39 has nothing to do with authors claims.

Line 214, ref. 42 has also nothing to do with authors writing.

Line 237, ref. 51 the same problem, no connection to manuscript.

Line 250, ref 53 is in Italian language, line 54 in French language, not even the titles are not in English, which doesn’t help much reviewers and readers.

Also, last paragraph in discussion section, lines 256 to 266 is not understandable at all what authors want to present and with very bad use of English language.

 Extensive editing of English language required, in many cases authors dont use correct english grammar but possible translate wrongly, like the use of the expression "with charge of 108 CFU/ml" or "the charge of E.coli". Also some instances in the text where english need to be checked as no clear meaning is possible.

Author Response

Dear Reviewer 4,

thank you for your review. The manuscript has been modified according to your comments and suggestions of the other reviewers. Below, I reported the replies, step by step. The changes according to your comments are coloured in green in the manuscript. References were renumbered in the manuscript, according to the changes in the paper.

Best regards

Elena Circella

This is the review of the manuscript “Antimicrobial Efficacy Of Allyl-Sulfide Against Escherichia Coli Strains From Rabbits “.

In general the use of bioactive components against infectious agents for human and animals are in center of research for many reasons. Unfortunately this manuscript has many flaws and it is written not with the proper attention.

Comment. To start with author’s credentials should be written in English, there must be an English translation of the 2nd institute.

Reply. The translation of the 2nd affiliation has been provided.

Comment. Simple summary and abstract, which is the 1st readers will check are very badly written, English editing and many mistakes, not clear scientific meanings and controversial results.

Reply. Simple summary and abstract have been changed providing the English language revision by MDPI Language Editing Services. The whole manuscript has been revised for English language.

Comment. In Introduction, very extensive introduction, to some points just narrative without any benefit and no use of references, i.e line 90-107.

Reply. According to your suggestion, some parts of Introduction have been deleted. Other ones have been added based on the other reviewers’ comments. In addition, other references have been provided.

Comment. In lines 68-70, the use of 1989 reference is quite old to talk about 20-30% losses plus, the reference used is not talking about due to E. coli the losses as authors claim.

Reply. 20-30% is the mortality caused by colibacillosis reported in that reference. We added a more recent reference about mortality rate.

Comment. In materials and methods section, authors claim that from 19 farms were taken samples of rabbits died from enteritis, without any other data of farms and no strains presented and no data of rabbits died i.e ages, weaned or not weaned at least.

Reply. Thank you for this comment that led us to explain more clearly the origin of bacterial strains and to provide additional information about rabbits and farms. The text of the manuscript has been changed accordingly.

Comment. In 2.4 paragraph, no references was presented for the method used for MIC and paragraph 2.5 for MBC.

Reply. We performed the tests according to CLSI and NCLSS, respectively (references were added to the manuscript) with some modifications, as described in detail. Specific guidelines are not available to test natural substances. Therefore, modifications to already known protocols and new methods are used for studies on antimicrobial activity of natural substances. We previously published a paper about antimicrobial efficacy of garlic against Salmonella Enteritidis strains, using a new method for MIC determination (Circella E., Casalino G., D’Amico F., Pugliese N., Dimuccio M.M., Camarda A., Bozzo G. In Vitro Antimicrobial Effectiveness Tests Using Garlic (Allium sativum) against Salmonella enterica Subspecies enterica Serovar Enteritidis. Antibiotics 2022, 11, 1481. https://doi.org/10.3390/antibiotics11111481).

Comment. The presence of table 2. In the results section, confuses readers as suddenly from nowhere we have a discussion about farms and strains but no mention of these was made earlier.

Reply. Thank you for your comment. The meaning of table 2 (reorganized according to the suggestion of reviewer 3) is clearer with the additional information reported in Material and methods section according to your suggestion. In fact, despite the similarity of the holdings and rabbits from which they came, an individual variability of strain susceptibility to the product was found.

Comment. In discussion paragraph there many problems regarding authors discussion about their results and the references used to back authors claims.

Line 201, ref. 35 is not supporting authors writing, also line 202, ref 36.

Line 205, ref. 39 has nothing to do with authors claims.

Line 214, ref. 42 has also nothing to do with authors writing.

Line 237, ref. 51 the same problem, no connection to manuscript.

Reply. Discussion paragraph has been checked more carefully, deleting some references mentioned accidentally and adding other more appropriate ones.

Comment. Line 250, ref 53 is in Italian language, line 54 in French language, not even the titles are not in English, which doesn’t help much reviewers and readers.

Reply. Although in Italian and French languages, we think that those references should be useful to the manuscript, and we would like not remove them if you agree.

Comment. Also, last paragraph in discussion section, lines 256 to 266 is not understandable at all what authors want to present and with very bad use of English language.

Reply. As above reported, the revision of English language has been provided by MDPI Language Editing Services.

Round 2

Reviewer 1 Report

This resubmission is a significant improvement over the original submission and meets my criteria for acceptance for publication. While now acceptable I am still of the opinion that the presentation of the data results in the loss of part of the data collected. However, I agree with the authors that alternative forms of presentation could be very difficult to implement.

Reviewer 2 Report

The paper can be published as is

The text appears correct

Reviewer 4 Report

no comments to be added

no comments to be added